# Single-cell map of dynamic cellular microenvironment of radiation-induced intestinal injury

Hao Lu [1,5], Hua Yan[1,5], Xiaoyu Li[1,5], Yuan Xing[1], Yumeng Ye[1], Siao Jiang[1,2], Luyu Ma[1], Jie Ping[1], Hongyan Zuo[1], Yanhui Hao[1], Chao Yu[1], Yang Li [1,3 ✉], Gangqiao Zhou [1,4 ✉] & Yiming Lu [1,2 ✉]

Intestine is a highly radiation-sensitive organ that could be injured during the radiotherapy for pelvic, abdominal, and retroperitoneal tumors. However, the dynamic change of the intestinal microenvironment related to radiation-induced intestine injury (RIII) is still unclear. Using single-cell RNA sequencing, we pictured a dynamic landscape of the intestinal microenvironment during RIII and regeneration. We showed that the various cell types of intestine exhibited heterogeneous radiosensitivities. We revealed the distinct dynamic patterns of three subtypes of intestinal stem cells (ISCs), and the cellular trajectory analysis suggested a complex interconversion pattern among them. For the immune cells, we found that Ly6c[+] monocytes can give rise to both pro-inflammatory macrophages and resident macrophages after RIII. Through cellular communication analysis, we identified a positive feedback loop between the macrophages and endothelial cells, which could amplify the inflammatory response induced by radiation. Besides, we identified different T cell subtypes and revealed their role in immunomodulation during the early stage of RIII through inflammation and defense response relevant signaling pathways. Overall, our study provides a valuable single-cell map of the multicellular dynamics during RIII and regeneration, which may facilitate the understanding of the mechanism of RIII.

[1] Beijing Institute of Radiation Medicine, Beijing 100850, China. [2] College of Life Sciences, Hebei University, Baoding City, Hebei Province 071002, China. [3] Academy of Life Sciences, Anhui Medical University, Hefei City, Anhui Province 230032, China. [4] Collaborative Innovation Center for Personalized Cancer Medicine, Center for Global Health, School of Public Health, Nanjing Medical University, Nanjing City, Jiangsu Province 211166, China. [5] These authors contributed equally: Hao Lu, Hua Yan, Xiaoyu Li. ✉email: leeyoung109@hotmail.com; zhougq114@126.com; ylu.phd@gmail.com

As one of the most sensitive organs to ionizing radiation, intestine is extremely susceptible to damage during the radiotherapy for pelvic, abdominal, and retroperitoneal tumors, which would lead to radiation-induced intestinal injury (RIII). Treatments of RIII are very limited and generally focused on reducing symptoms, and the curative effects are not satisfactory. RIII is largely defined as clonogenic cell death and apoptosis in the crypt cells, which results in insufficient replacement of villus epithelium, breakdown of the mucosal barrier[1,2], inflammation and immune abnormality[3]. Previous studies mainly focused on the roles of molecules and pathways in DNA damage, apoptosis, autophagy[4], inflammation and immune[5]. However, the dynamics of the microenvironment during intestine injury and regeneration remains poorly understood, which limits the elucidation and treatment of RIII. Therefore, the present study aims to clarify the dynamic variation of the cellular microenvironment of RIII.

It has been widely accepted that the intestinal stem cells (ISCs) are able to replenish the whole crypt–villus axis, generating all differentiated cell types required for the physiological function of the intestine[6]. A number of ISC subpopulations in the small intestine have been identified, including Lgr5+ crypt-based columnar cells (CBCs), +4 reserve stem cells (RSCs) and revival stem cells (revSCs). Lgr5+ CBCs are considered indispensable for intestine recovery following exposure to radiation[7]. +4 RSCs, which express specific markers *Bmi1*, *Hopx* and *Tert*[8–10], have been described as a slow dividing reserve stem cell population. Recently, a group of revSCs was identified to be extremely rare under homeostatic conditions and arise in damaged intestines to reconstitute Lgr5+ ISCs and regenerate the intestine[11]. Irradiation causes a sharp reduction in the number of ISCs, which brings great challenges in the repair of injured intestinal epithelium. Despite the advances in our understanding of the ISCs in the past few years, a dynamic landscape of ISCs during injury and regeneration is still lacking and their interconversion relationships remain puzzling.

Immune cells also play important roles in the pathogenesis of RIII. Macrophages are crucial component of the immune system in the intestine, which modulate inflammation by secreting distinct cytokines and acting as professional phagocytes[12,13]. Intestinal macrophages require continuous replenishment by blood monocytes and have very poor proliferative capacity, which is different from the other tissue macrophages. Exposure to radiation can significantly decreases the levels of macrophages in the damaged intestine[14]. Besides, the lymphocytes, particularly T cells, also participate in the immune regulation through direct killing the infected cells or recruiting other immune-protective and regulatory cells[15]. Increased infiltration of activated T cells have been observed in colonic mucosa post irradiation[16]. However, the heterogeneity of T cell subtypes, as well as their radiosensitivity, were still not well deconvolved. Furthermore, the cross-talk between immune cells and other cells is also of great significance for the initial inflammatory response and microenvironment homeostasis recovery post irradiation and remains to be elucidated.

Recently, the single-cell RNA sequencing (scRNA-seq) has been applied to identify new cell types or cell states[17,18], investigate cellular plasticity and stemness of ISCs[11,19], or trace developmental relationships among different cell populations in the intestine[20]. However, these studies have not investigated the dynamic changes of microenvironment during intestine injury and regeneration. Here, we utilized the scRNA-seq to explore the multicellular relationship of homeostatic and regenerating intestine in a time-course manner. We generated transcriptomes of 22,680 single cells in the intestinal microenvironment, trying to profile the dynamics of ISCs and immune cells located in the mucous layer. Our data can be a valuable resource for further investigation on the cellular mechanism of RIII and development of potential therapy strategies.

## Results

**A dynamic single-cell map of cellular microenvironment in healthy and injured small intestine.** Mice were exposed to 15 Gy of abdominal irradiation using a $^{60}$Co irradiator to induce intestinal injury (Fig. 1a). About 20% mice died in the irradiation group within 3–6 days post irradiation (Fig. 1b). The body weights of mice after irradiation exposure continued to lose in the first 7 days and then gradually increased afterward (Fig. 1c). In line with this, exposed mice showed dramatically reduced intestinal weight at day 3 and day 7 post irradiation as compared to unexposed mice and showed recovery at day 14. Morphological and cellular phenotype analyses also showed significant decrease of villus length, villus width, crypt depth and number of crypts at day 1 and/or day 3 post irradiation (Supplementary Fig. 1a–e). A peak of TUNEL-positive cells at day 1 post irradiation suggested that the radiation-induced cell death occurs mostly within the first 24–48 h (Supplementary Fig. 1f). An increase of Ki-67-positive cells from day 3 to day 14 indicated that the day 3–14 is a key time-window for intestinal regeneration (Supplementary Fig. 1g).

To generate a dynamic single-cell map of the intestinal microenvironment related to RIII, we employed a droplet-based scRNA-seq approach to profile the transcriptomes of single cells from the intestinal tissues at day 1 ($n = 4$), day 3 ($n = 3$), day 7 ($n = 3$) and day 14 ($n = 3$) after irradiation exposure as well as the unexposed healthy intestinal tissues ($n = 4$) (Fig. 1a). After quality filtering, we obtained the transcriptomes of a total of 22,680 single cells, with an average of 1778 genes and 7843 unique transcripts per cell (Supplementary Fig. 2a, b; Supplementary Table 1). These transcriptomes of single cells from all samples were merged using a canonical correlation analysis (CCA)-based batch correction approach to generate a global map of cellular microenvironment of healthy and injured intestines. Shared nearest neighbor (SNN) graph-based clustering of single cells identified a total of 54 cell clusters (subtypes), which were visualized on the *t*-distributed stochastic neighbor embedding (tSNE) dimensional reduction map (Fig. 1d and Supplementary Fig. 2c). Examination of the possibility of doublets revealed that the doublet rates in all clusters were remarkably low, remaining below 2% (Supplementary Fig. 2d). Differentially expressed genes were then calculated for each cell cluster using the Wilcoxon rank sum test.

Using canonical marker genes, we identified ten major cell types in our dataset, including ISCs, transit amplifying cells, enterocytes, goblet cells, enteroendocrine cells (EECs), Paneth cells, endothelial cells, T cells, B cells and myeloid cells, and most of them consist of multiple subtypes (Fig. 1e and Supplementary Fig. 2e), suggesting a complex cellular microenvironment of healthy and irradiation-injured intestines. To test the robustness and reliability of the annotation result, we re-analyzed the data with the updated version of Cell Ranger and Seurat packages (see Methods) and found that cell types from the updated versions agree well with those from the current ones (Supplementary Fig. 3). The intestinal microenvironment changed greatly during RIII and regeneration (Fig. 1f, g). Specifically, the stem cells and immune cells (T cells, B cells and myeloid cells) decreased markedly at day 1 post irradiation; enterocytes decreased sharply at day 3; and immune cells and endothelial cells exhibited a dramatic increase at day 3. Despite of the great alterations of intestinal microenvironments during the first 7 days post irradiation, intestinal tissues at day 14 exhibited very similar

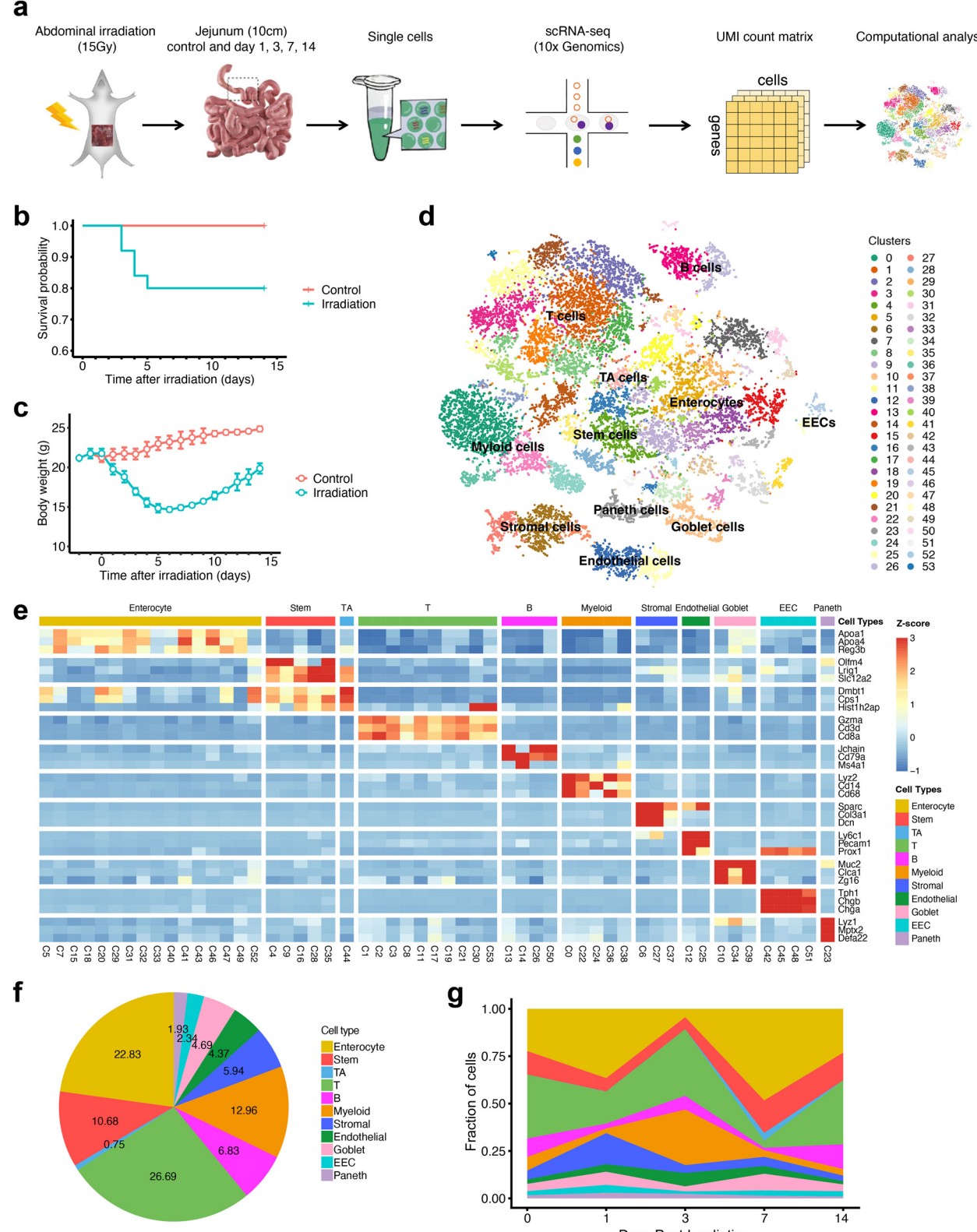

**Fig. 1 Identification of major intestinal cell types and their markers using scRNA-seq. a** Overview of single-cell RNA sequencing (scRNA-seq) analysis for the irradiation-induced intestinal injury (RIII). **b** Survival rates of the mice exposed to 15 Gy abdominal irradiation and in control group. **c** Body weights of the mice exposed to 15 Gy abdominal irradiation and in control group. The error bars represent the standard deviation (SD). **d** tSNE projection of the 22,680 cells profiled, colored by Seurat cluster and annotated with major cell types. **e** Heatmap displaying the z-score normalized mean expression of cell type-specific canonical marker genes across clusters. **f** Pie chart of cell type fractions in all sequenced samples. **g** Area chart showing the dynamic changes of the proportions of major cell types in healthy intestinal samples (Control) and exposed intestinal samples at different times after irradiation. D1, day 1; D3, day 3; D7, day 7; and D14, day 14.

cellular composition with those non-irradiated ones (Fig. 1g, Supplementary Fig. 4). These results suggested that intestinal cellular microenvironment disrupted by intense abdominal irradiation can be largely reconstructed within 14 days post irradiation, which is in line with our morphological observations (Supplementary Fig. 1).

**Intestinal cellular microenvironment showed heterogeneity of in vivo radiosensitivities**. The dynamic change of cellular composition in intestines before and after irradiation exposure provided an opportunity to explore the in vivo radiosensitivities of various cell subtypes. To quantify the radiosensitivity for each cell cluster, we compared the ratios between the observed and expected cell numbers from the intestinal samples at day 1 after irradiation. Consistent with previous studies[7,21], most ISC and immune cell subtypes are highly radiosensitive, exhibiting significantly lower frequencies than expected at day 1 (Fig. 2a). Nevertheless, there are a few ISC and immune cell subtypes (C22, C30, C19, and C35) exhibited similar or higher frequencies than expected at day 1, suggesting they are radioresistant subpopulations. Apart from ISCs and immune cells, we found all the endothelial and stromal cell subtypes are radioresistant, while the enterocytes, goblet cells and EECs exhibit heterogeneous levels of radiosensitivity across their respective subtypes.

We next sought to identify the pathways activated by various cell types to confront irradiation-induced apoptosis. We focused on the top four enriched major cell types in our dataset, including the ISCs, enterocytes, myeloid cells and T cells. For each cell type, we identified pathways significantly up-/down-regulated in surviving cells at day 1 post irradiation using the whole cells from non-irradiated samples as background (Fig. 2b, c and Supplementary Fig. 5). Notably, a number of pathways, including PI3K/AKT/mTOR signaling, MYC signaling, TGF-β signaling and cell cycle-related pathways, were consistently downregulated in surviving cells of different cell types, while KRAS-down signaling pathway were consistently upregulated in surviving cells. We also investigate the genes that are upregulated in survived cells at day 1 post irradiation as compared to non-irradiated cells for the stem cells, enterocytes, myeloid cells and T cells, respectively. We identified 67 significantly upregulated genes that were shared by all the major cell types (Fig. 2d and Supplementary Table 2). Functional annotation showed that the upregulated genes include several genes known to be involved in defense response and negative regulation of apoptotic process (Fig. 2e and Supplementary Table 3).

**Dynamics of three distinct ISC subpopulations during intestinal injury and regeneration**. ISCs are critical for epithelium regeneration after injury[7,11]; however, their phenotypic heterogeneity and dynamics during intestinal injury and regeneration have not been fully characterized. We identified a total of 2411 ISCs that were divided into five clusters (Fig. 3a, b). Seminal studies defined Lgr5 as a molecular marker of stem cell residing in the base of crypt, referred to as Lgr5+ CBCs[22]. Due to the limited sequencing sensitivity for *Lgr5* mRNAs, we didn't detected enough expression of *Lgr5* in the stem cells, as shown in the previous studies[18,23]. We found that other Lgr5+ CBC markers, including *Olfm4*, *Smoc2*, and *Prom1*, were highly expressed in clusters C4, C9, and C16, indicating they are Lgr5+ CBCs; stem cells in C35 expressed high levels of +4 RSC signatures (*Hopx* and *Tert*), suggesting they are a set of +4 RSCs[8,10]. Cluster C28 was enriched for stem cells that highly expressed *Clu*, *Cxadr* and *Anxa1*, which are signatures of recently reported Clu+ revSCs[11].

We then investigated the temporal dynamics of these ISC subpopulations. The proportion of Lgr5+ CBCs was drastically

reduced at day 1 and day 3 and was substantially recovered at day 14 post irradiation (Fig. 3c). In line with the scRNA-seq data, the integral luminous density of in situ hybridization with probes specific for *Olfm4* showed similar results (Fig. 3d, e). By contrast, +4 RSCs in C35 were specifically enriched at day 1 post irradiation, suggesting they are not only radioresistant but can also be induced by irradiation (Fig. 3c), in line with a previous study that detected increased lineage tracing output of +4 RSCs after irradiation exposure[24]. Distinct from Lgr5+ CBCs and +4 RSCs, Clu+ revSCs in C28 were specifically enriched at day 3 post irradiation (Fig. 3c), in accordance with the study that first reported the population of Clu+ revSCs[11]. Collectively, these results showed the distinct dynamic patterns of three ISCs subpopulations. Moreover, the temporal sequential enrichment of +4 RSCs, Clu+ revSCs and Lgr5+ CBCs during this process hints at a potential sequential differentiation relationship among them.

**Differentiation trajectory analyses of intestinal stem cell subpopulations**. We next sought to explore the differentiation trajectories among these heterogeneous ISC subpopulations. RNA velocity has recently emerged as a powerful approach for inferring the transition direction of a single cell to neighboring cells[25]. Here, we performed RNA velocity analysis on stem cell populations in a time-course manner. Given that only +4 RSCs were enriched at day 1, we combined them with ISCs presented under homeostatic conditions (Control-D1) or those presented at day 3 (D1-D3), respectively, to investigate the inter-subpopulation differentiation trajectories. Notably, the Control-D1 RNA velocity map showed that there is a velocity flow from Lgr5+ CBCs toward +4 RSCs (Fig. 3f), in accordance with a previous study that demonstrated the interconversion between Lgr5+ CBCs and +4 RSCs[10]. This result may explain the specific enrichment of +4 RSCs at day 1 post irradiation. We observed two clear velocity flows in the D1-D3 map (Fig. 3g). The first flow initiates from the +4 RSCs toward Clu+ revSCs, suggesting +4 RSCs may be able to differentiate to Clu+ revSCs. This result is not only in accordance with the specific enrichment of Clu+ revSCs at day 3 post irradiation rather than at day 1 (Fig. 3c), but also consistent with the observation of a peak of CLU-tdTomato+ clone frequency at the +4 position from the crypt bottom[11], where +4 RSCs are specifically enriched. The second flow is from the Clu+ revSCs toward Lgr5+ CBCs, suggesting that Clu+ revSCs could repopulate Lgr5+ CBCs when the latter were almost eliminated by irradiation, in line with the previous study[11]. These results revealed the potential relationships among the ISC subtypes, though further mechanism experiments are required to test and verify the specific interconversion process.

We then used SCENIC[26] to investigate the potential key regulons along these velocity flows. We found Myc and Hdac1 were among the top transcription factors (TFs) that exhibit remarkably increasing activities along the velocity flow from Lgr5+ CBCs toward +4 RSCs (Fig. 3h). Myc has been reported to be required for intestinal formation and regeneration but dispensable for homeostasis of the adult intestinal epithelium[27,28]. Here, our result suggested Myc may facilitate intestinal regeneration by promoting the conversion from Lgr5+ CBCs to +4 RSCs upon radiation-induced injury. Besides, Hdac1 was also reported to regulate ISC homeostasis[29]. Among the top correlated TFs along the velocity flow from +4 RSCs toward Clu+ revSCs (Fig. 3i), Ybx1 is a stress-activated TF and has been reported to be able to transcriptionally activate *Clu* expression by directly binds to the promoter regions of *Clu*[30]. Our data suggest Ybx1 may play important roles in the induction of Clu+ revSCs from +4 RSCs. Besides, another TF Hmga1 has been reported to amplify the Wnt signaling and expand the ISC compartment and Paneth cell

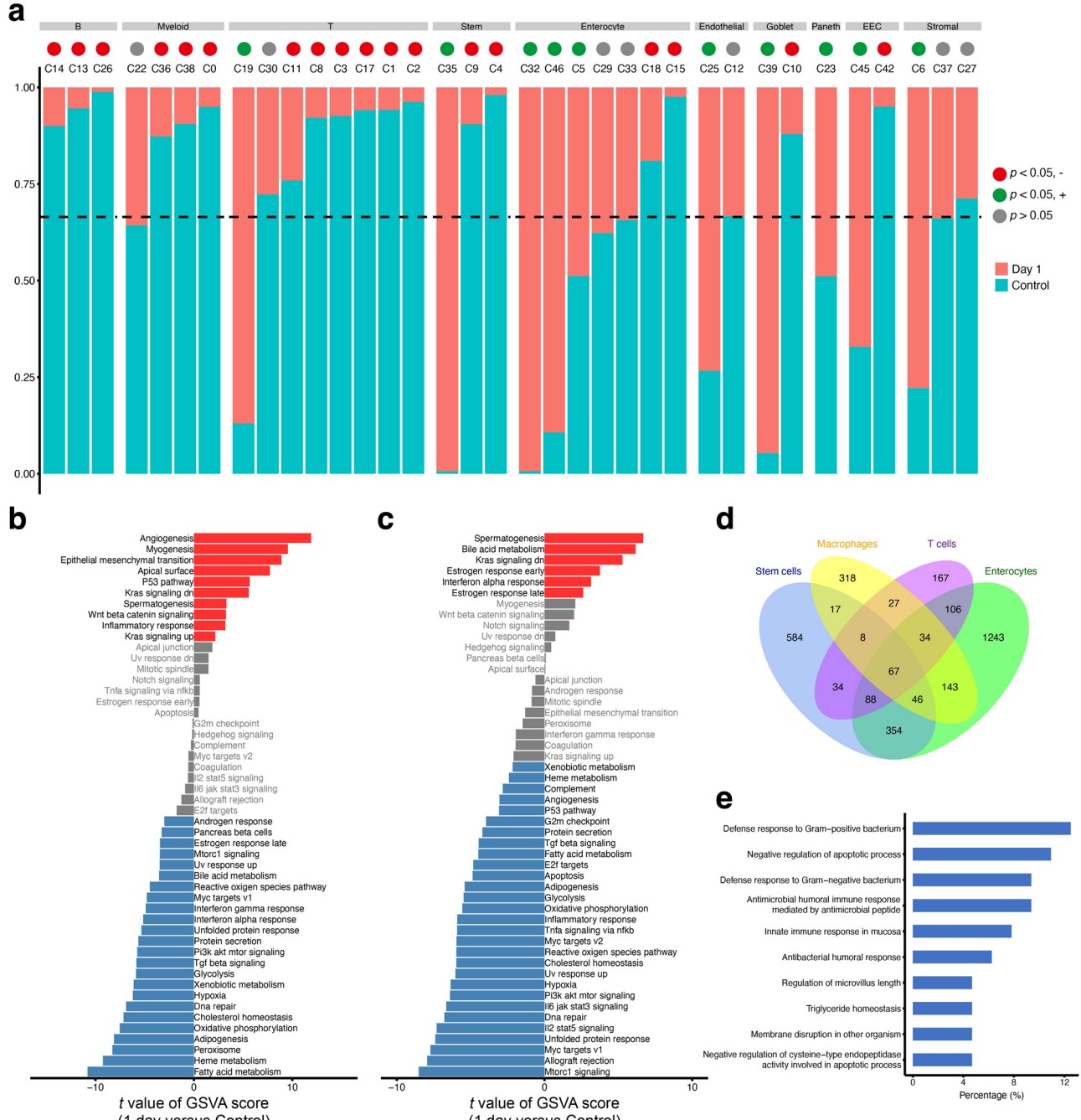

**Fig. 2 Characteristics of in vivo radiosensitivities of different cell types in intestine. a** Bar plots showing the fraction of cells originating from the control and day 1 groups in each cluster. Only the clusters that had >50 cells from the two groups and that showed no inter-individual difference (clusters with no more than 70% of cells from a single sample) were shown. Vertical dashed lines indicate the overall fraction of cells originating from the control group in all shown clusters. **b** Difference of hallmark pathway activities between stem cells from the control and day 1 (D1) groups. Shown are *t* values calculated in a linear modal comparing the pathway scores estimated by gene set variation analysis (GSVA) between cells from the two groups. **c** The same as (**b**) for macrophages from the control and day 1 groups. **d** Venn diagram showing the intersection of differentially expressed genes between cells from the control and day 1 groups for stem cells, macrophages, T cells, and enterocytes. **e** Top 10 significantly enriched gene ontology (GO) terms of 67 common differentially expressed genes for stem cells, macrophages, T cells, and enterocytes.

niche[31]. For the velocity flow from Clu[+] revSCs toward Lgr5[+] CBCs, we found Runx1 and Stat3 were among the top correlated TFs (Fig. 3j). Stat3 has been shown to be indispensable for damage-induced crypt regeneration[32] and the Runx1-Stat3 signaling pathway has been reported to regulate the differentiation of Lgr5[+] CBC[33]. Taken together, our results revealed the possible differentiation trajectories among Lgr5[+] CBCs, +4 RSCs

and Clu[+] revSCs upon intestinal injury, and identified the specific TF sets driving these differentiation paths.

**Bidirectional differentiation of peripheral monocytes into pro-inflammatory macrophages and resident macrophages.** A total of 4010 myeloid cells were identified in our dataset, which

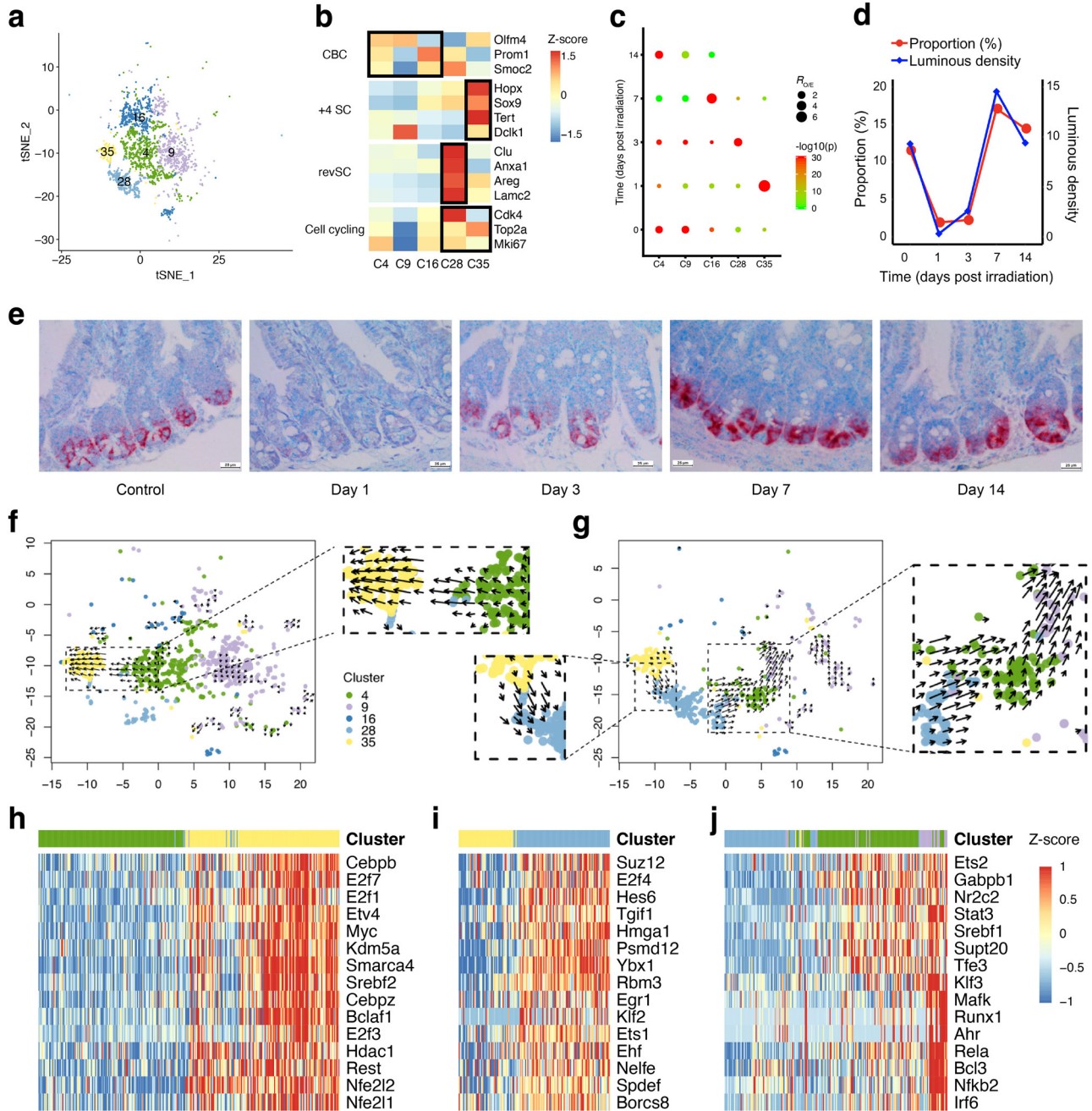

**Fig. 3 Dynamics and differentiation trajectories of intestinal stem cell (ISC) subsets in irradiation-induced intestinal injury and regeneration. a** tSNE projection of 2,411 ISCs identified, colored by Seurat cluster identities. **b** Heatmap of the average expression of the selected ISC function-related marker genes in five ISC clusters. **c** Dot plots showing the ratio of observed to expected cell numbers ($R_{O/E}$) of each ISC cluster in the indicated samples, with log-transformed Bonferroni-corrected $P$ values in Chi-square tests. **d** Line charts showing the average luminous density of *Olfm4* in immunostaining (blue) and the fraction of *Olfm4*+ CBCs (clusters 4, 9 and 16; red) in the indicated groups. **e** *Olfm4* immunostaining (red) in jejunum in the control, day 1, 3, 7 and 14 groups. Scale bar = 25 μm. **f** RNA velocities of ISCs from the control and day 1 groups visualized on the tSNE projection. **g** The same as (**f**) for ISCs from the day 1 and day 3 groups. **h** Heatmap depicting the estimated activity of top 15 regulons showing differential activation in ISCs along the velocity flow from the control and day 1 groups, which is depicted by a black dashed line in (**f**). Shown are normalized mean area under the curve (AUC) scores of expression regulation by each transcription factor estimated in SCENIC. Cells are ordered according to first principal component (PC1) coordinate to grasp the primary velocity orientation. **i**, **j** The same as (**h**) for ISCs along the velocity flow from the day 1 and day 3 groups, which is depicted by black dashed lines in the left (**i**) and right (**j**) panel in (**g**).

were divided into five clusters (C0, C22, C24, C36, and C38) (Fig. 4a). Cells in cluster C0 expressed high levels of monocyte markers (*Cd14*, *Ccr2*, and *Ly6c2*), indicating they are peripheral Ly6c+ monocytes. Cells in C22 highly expressed macrophage markers (*Cd80*, *Cd206*, *Cd64* and *F4/80*) and a set of pro-inflammatory genes (*Il1a*, *Il1b*, *Il6* and *Tnf*), suggesting they are a

set of pro-inflammatory macrophages. C36 was enriched for cells expressed high levels of resident macrophage markers (*Cx3cr1* and *Fcgr1*), suggesting they are Cx3cr1+ resident macrophages. Besides, we also identified a cluster of neutrophils (C24) highly expressing neutrophil markers (*Ly6g*, *Retnlg* and *S100a9*) (Fig. 4b).

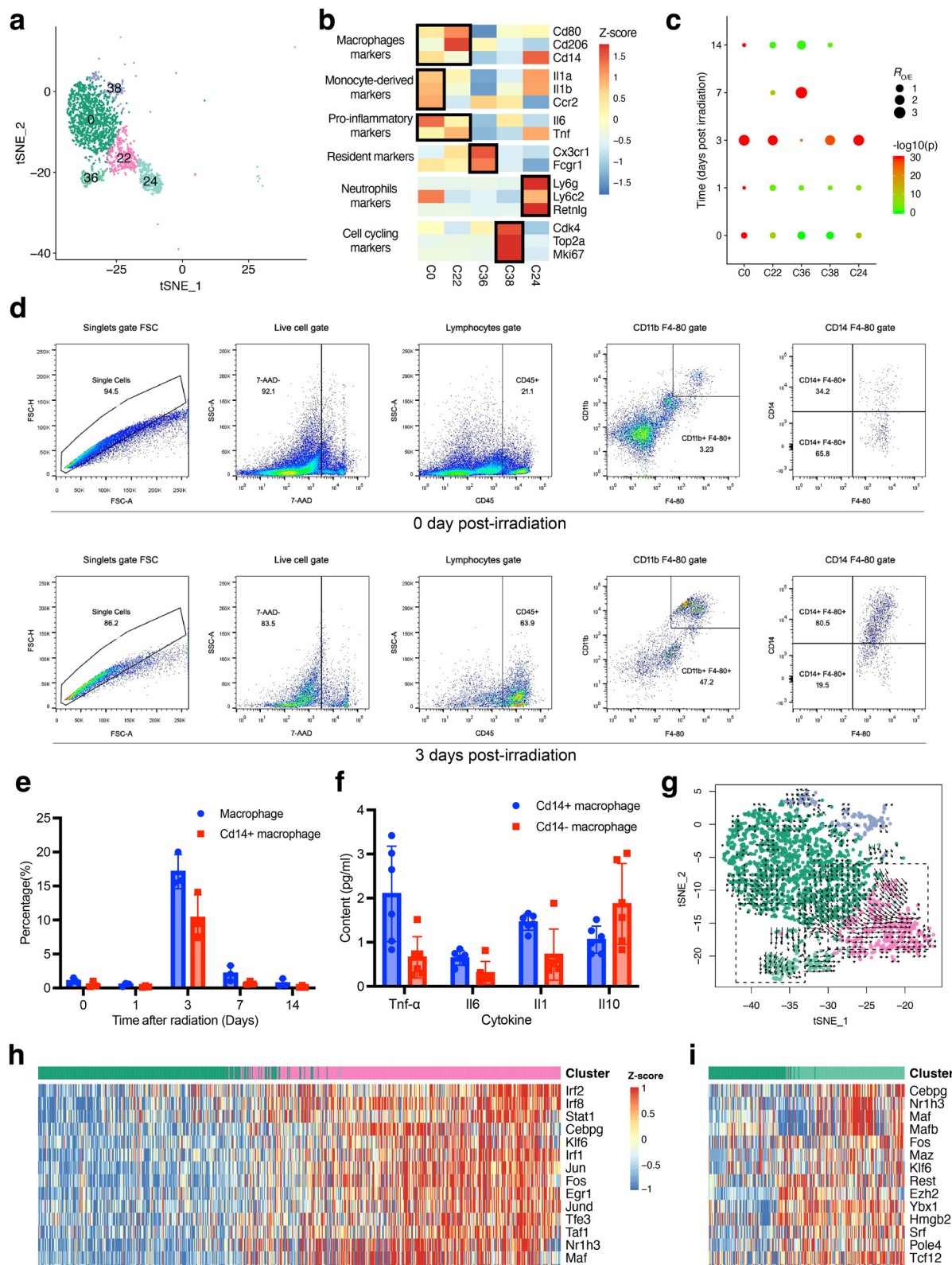

We then investigated the temporal distribution of these myeloid subpopulations. We found that Ly6c$^+$ monocytes (C0), pro-inflammatory macrophages (C22) and the neutrophils (C24) were highly enriched at day 3 post irradiation (Fig. 4c). The temporal distributions of pro-inflammatory macrophages after irradiation were confirmed by flow cytometry, and their expression of pro-inflammatory cytokines were measured

(Fig. 4d–f). In contrast, Cx3cr1$^+$ resident macrophages in C36 showed depletion at day 1 and day 3 post irradiation and became abundant at day 7, suggesting resident macrophages are radio-sensitive and could rapidly recover after irradiation. However, additional validation of this radiosensitivity disparity among different myeloid subpopulations necessitates experiments involving whole-body irradiation since non-irradiated Ly6c$^+$

**Fig. 4 The characteristics and roles of macrophage subsets in the irradiation-induced intestinal inflammation. a** tSNE projection of 2,926 myeloid cells identified, colored by Seurat cluster identities. **b** Heatmap of the average expression of the selected myeloid cell function-related marker genes in five myeloid cell clusters. **c** Dot plots showing the ratio of observed to expected cell numbers of each myeloid cell cluster in the indicated samples, with Bonferroni-corrected *P* values by Chi-square tests. **d** Flow cytometry analyses showing the percentages of Cd14+ inflammatory macrophages at control and 3 days post irradiation groups. **e** Dynamic change of the percentages of Cd14+ macrophages at control, 1, 3, 7 and 14 days post irradiation groups. The error bars represent the SD. **f** Expression levels of inflammatory cytokines by Cd14+ and Cd14- macrophages in intestine 3 day post irradiation. The error bars represent the SD. **g** RNA velocities of four macrophage clusters visualized on the tSNE projection. **h, i** Heatmap depicting the estimated activities of top 15 differentially activated regulons along the velocity flow, which was depicted by black dashed lines in the right (**h**) and left (**i**) panel in (**g**). Shown are normalized mean area under the curve (AUC) scores of expression regulation of each transcription factor estimated by SCENIC. Cells are ordered according to first principal component (PC1) coordinate to grasp the primary velocity orientation.

inflammatory cells would be recruited from circulation in response to abdominal irradiation.

Using RNA velocity analysis, we found that there are two differentiation paths initiated from Ly6c+ monocytes towards pro-inflammatory macrophages and Cx3cr1+ resident macrophages, respectively (Fig. 4g). Although the alternative outcomes of Ly6c+ monocytes to be resident and pro-inflammatory macrophages has been previously reported in inflammatory bowel disease (IBD)[34], our result shows that this bidirectional differentiation pattern may be import for RIII recovery. Further, SCENIC analysis was performed to investigate the potential key regulons driving these two differentiation paths, respectively. We found these two differentiation paths shared several top correlated TFs, including Cebp, Maf, Mafb and Nr1h3 (Fig. 4h, i). CCAAT/enhancer-binding protein (Cebp), Maf and Mafb TFs has been reported to be important for monocyte-to-macrophage differentiation[35]. Nuclear receptor Nr1h3 (LXRα) has also been reported to be a major regulator of macrophage development[36]. Besides the shared TFs, the two differentiation paths were also driven by specific factors. Activities of Jun, Stat1, Irf1 and Irf2 were specifically upregulated along the velocity flow from Ly6c+ monocytes to pro-inflammatory macrophages (Fig. 4h), while Myc and Myc-associated zinc finger (Maz) were specifically upregulated along the velocity flow from Ly6c+ monocytes to resident macrophages (Fig. 4i), in accordance with the key role of Myc in the self-renewal activity of macrophages[37]. Together, these results suggested that peripheral monocytes could differentiate into both pro-inflammatory macrophages and resident macrophages after RIII, and these two differentiation paths may be driven by different key TFs.

**A positive feedback loop between the macrophages and endothelial cells amplifies inflammatory response upon RIII.** To assess the roles of various cell types in the inflammatory response induced by irradiation, we then investigated the cell-cell communication (CCC) in the intestine microenvironment. We found that the overall and the myeloid cell-derived interaction strength showed a sharp decrease in day 1 post irradiation and a remarkable increase in day 3 post irradiation (Fig. 5a). Given the significant enrichment of myeloid cells in day 3, we then compared the CCC between day 3 and day 1 in more details. Results revealed that the interaction number and strength among myeloid, endothelial and stromal cells exhibited strong upregulation in day 3 compared to day1 (Fig. 5b). It is, moreover, noteworthy that myeloid cells showed markedly increased cellular interaction in signaling pathways relevant to leukocyte migration and adhesion, such as ICAM, ITGAL-ITGB2, SELPLG, and SELL signaling (Fig. 5c and Supplementary Fig. 6). It is reasonable to postulate that the high-volume infiltration of myeloid cells in day 3 might be induced by endothelial and stromal cells.

We then examined the CCC associated with different myeloid cell populations. We found that ICAM and ITGAL-ITGB2 signaling pathways exhibited remarkable enrichment between endothelial cells and monocytes and pro-inflammatory

macrophages (Fig. 5d). Investigation of specific ligands and receptors revealed that endothelial cells expressed high levels of adhesion molecules such as *Icam1* and *Icam2*, to recruit monocytes and pro-inflammatory macrophages (Fig. 5e, f), in accordance with previous studies[38]. Besides the adhesion molecules, we found these endothelial cells also expressed high levels of chemokine *Ccl5* and *Cxcl12* to recruit pro-inflammatory macrophages (Fig. 5e, f). Interestingly, the pro-inflammatory macrophages expressed high levels of vascular endothelial growth factor (*Vegf*), which could, in turn, enhance the endothelial cell survival and proliferation[39] (Fig. 5e, f). The interactions between the activated endothelial cells and pro-inflammatory macrophages could amplify the proliferation of both populations, which is consistent with the specific enrichment of pro-inflammatory macrophages and endothelial cells at day 3 post irradiation. We speculate the amplification effect is critical for the quick induction of inflammation at early stage of intestinal injury.

We also sought to identify the pathways that are upregulated upon endothelial cells activation during RIII. Compared to endothelial cells in homeostatic intestine, the activated endothelial cells at day 3 post irradiation were shown to be upregulated in inflammatory response, TNF-α signaling via NF-κB, EMT and KRAS signaling pathways (Supplementary Fig. 7). NF-κB has been reported to play a pivotal role in the inducible expression of cytokines in inflammatory response induced by irradiation[40]. Besides, we found KRAS signaling pathway may also play an important role in inflammatory cytokines induction. In addition, the upregulation of EMT pathway indicates that irradiation could induce the epithelial-to-mesenchymal transition of endothelial cells, which may be involved in intestinal fibrosis after radiation exposure[41]. Collectively, these results showed that the cellular interactions between macrophages and endothelial cells could achieve a quick amplification of inflammation upon the RIII.

**Dynamics of T cells in RIII and recovery.** We detected 6027 T cells in our dataset, which were divided into ten clusters and further annotated as Cd4+ naïve T cells (TNs; C8), Cd8+ effector T cells (TEs; C1, C2, C17, and C19), Cd8+ tissue resident memory T cells (TRMs; C3, C11, and C21), and proliferative T cells (T.Prol; C30 and C53) (Fig. 6a, b). To characterize their functions in more details, Gene Ontology term enrichment was performed in each of four populations. Although both Cd8+ TEs and Cd8+ TRMs showed higher expression in cytotoxicity-related molecules, Cd8+ TEs were mainly involved in the regulation of cellular catabolic process and defense response to bacterium, while Cd8+ TRMs were mostly enriched in antioxidant activity and hemoglobin complex relevant pathways (Fig. 6c). Besides, Cd4+ TNs and proliferative cells were significantly involved in regulation of T cell activation and cell-cell adhesion, and mitotic cell cycle relevant pathways, respectively (Fig. 6c). Exploration of their temporal patterns showed that Cd8+ TRMs were enriched in the control group (Fig. 6d), in line with the observation that TRMs constituted the majority of T cells in normal intestine[42].

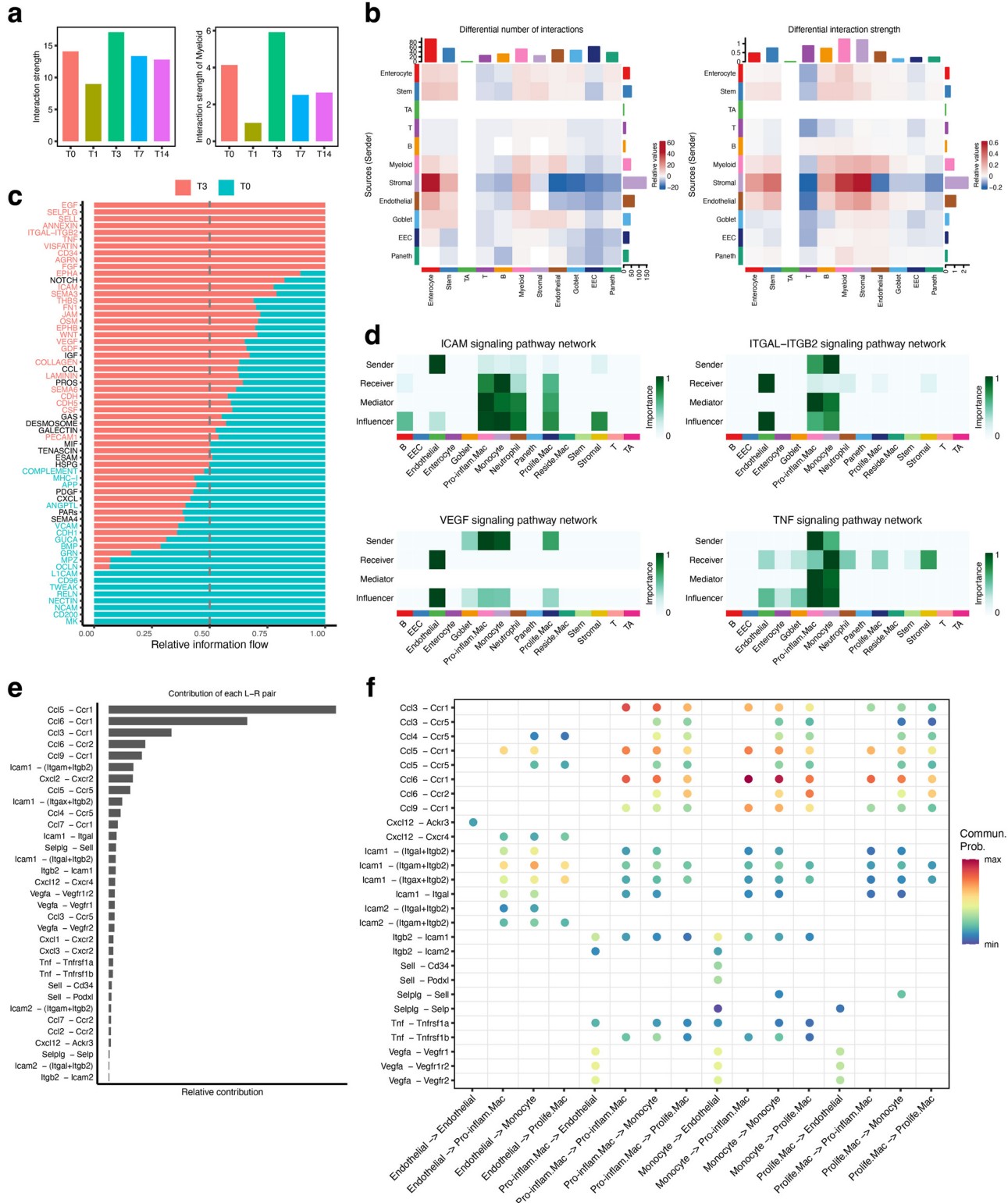

**Fig. 5 Cellular interaction between the myeloid and endothelial cells. a** The overall (lift) and myeloid cell-derived (right) interaction strength at different times post irradiation. **b** Heatmap depicting the differential interaction numbers (left) and strength (right) among major cell types in day 3 group compared to those in control group. **c** Bar graph showing the overall information flow of each signaling pathway in day 3 and control group. **d** Heatmap depicting the network centrality scores for each cell type in the selected signaling pathways. **e** Bar graph showing the contribution of each ligand-receptor pair to the myeloid cell migration and activation relevant signaling pathways. **f** Dot plot showing the significant ligand-receptor pairs associated with the myeloid cell migration and activation relevant signaling pathways.

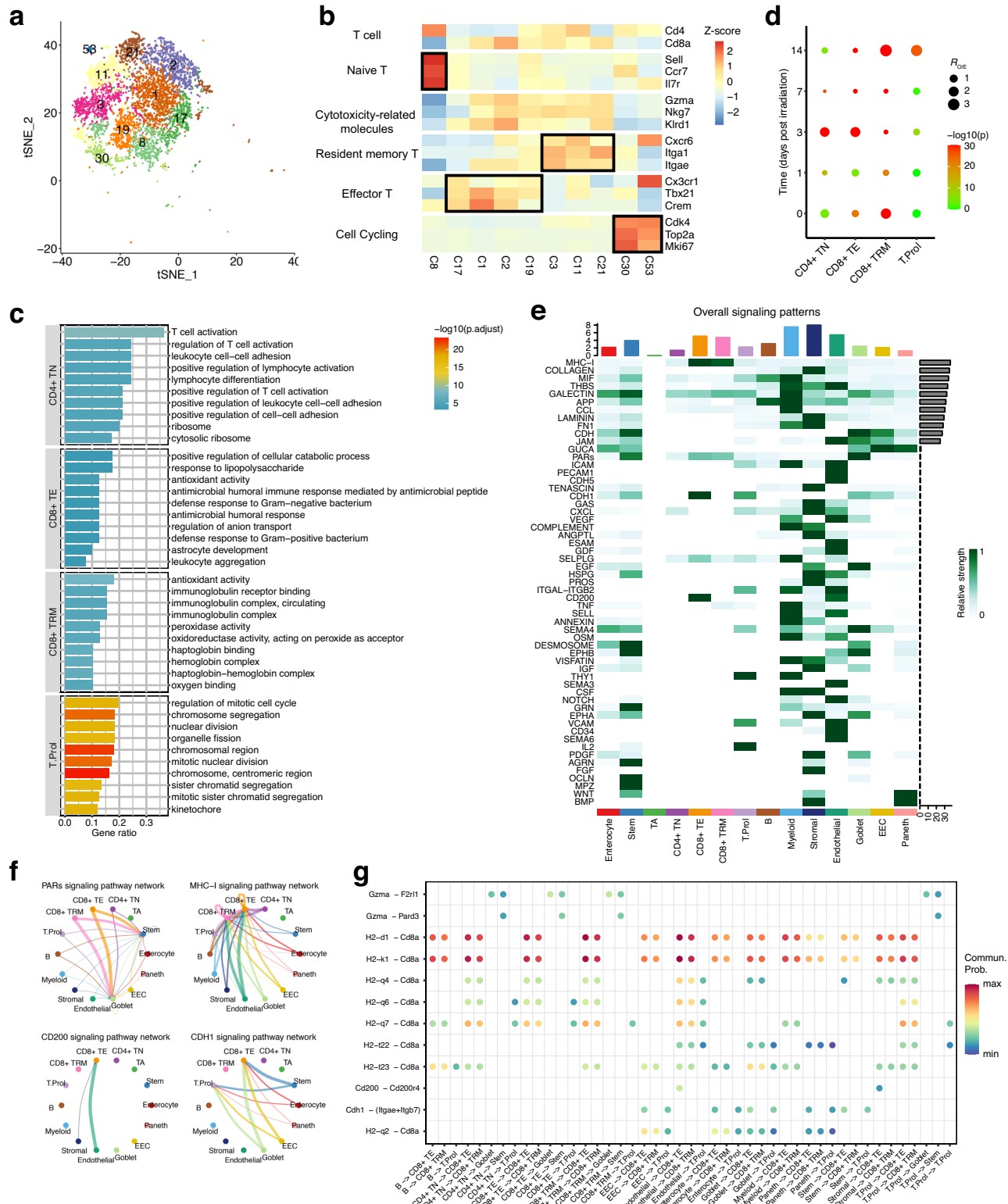

**Fig. 6 The characteristics of T cell subtypes and the associated cellular interaction in irradiation-induced intestinal inflammation. a** tSNE projection of 6027 T cells identified, colored by Seurat cluster identities. **b** Heatmap of the average expression of the selected T cell function-related marker genes in 10 T cell clusters. **c** The top 10 significantly GO terms of the marker genes in each T cell subtype. **d** Dot plots showing the ratio of observed to expected cell numbers of each T cell subtype in the indicated samples, with Bonferroni-corrected *P* values by Chi-square tests. **e** Heatmap showing the relative strength of overall signaling pathways associated with each cell population in the day 3 group. **f** Circle plot depicting the inferred signaling network of the selected signaling pathways in the day 3 group. **g** Dot plot showing the significant ligand-receptor pairs associated with the signaling pathways shown in (**f**).

This population were significantly reduced at days 1–7 post irradiation and not restored until day 14. In contrast, both Cd8+ TEs and Cd4+ TNs exhibited remarkable enrichment in day 3, suggestive of an important role in the early inflammation stage.

We next investigated CCC processes involving different T cell populations in day 3. We found that strong communication probabilities of immunomodulation and defense response relevant signaling pathways, including PARs[43], MHC-I[44], CD200[45], and CDH1[46] signaling pathways, were associated to different T cell subsets (Fig. 6e, f). Of note the Cd200-Cd200r4 interaction was specifically associated to the CD8+ TE cells with endothelial and stromal cells (Fig. 6g). CD200 is a transmembrane protein with known inflammation regulation properties. It has been reported that the engagement of CD200 to its receptor CD200R would induce a immunosuppressive phenotype on macrophages[47], mast cells[48], basophils[49], NK Cells[50], etc. These result lead us to postulate that endothelial and stromal cells would participate in the regulation of immune responses through CD200 signaling mediated immunosuppressive effect on CD8+ TEs.

## Discussion

The dynamics of cellular microenvironment during RIII and regeneration remains largely uncharacterized. In this study, by combining the single-cell transcriptome profiling with temporal distribution analysis, lineage reconstruction, TF profiling and cellular interaction analyses, we provided a comprehensive dynamic landscape of the cellular microenvironment during intestinal injury and regeneration.

The radiosensitivity of cells varies greatly, mainly depending on the degree of cell differentiation, cell proliferation ability, metabolic status and surrounding environment[3,51]. Currently, it is widely accepted that the lymphocytes, hematopoietic cells, small intestinal crypt cells and germ cells are highly radiosensitive cell types, but these kinds of cells are composed with multiple subtypes with different biological characteristics. In this study, based on scRNA-seq, we assessed the in vivo radiosensitivity of different cell subtypes in the intestinal microenvironment. Our data not only showed that diverse cell subtypes in the intestinal microenvironment exhibited highly heterogeneous levels of radiosensitivity, but also revealed cross-cell consistency of pathway activation for various cell types to confront the radiation-induced apoptosis.

Much attention has been focused on ISC injury and regeneration post irradiation[7,11,24,52]. It is generally accepted that Lgr5+ stem cells were sensitive to irradiation injury while +4 stem cells show relatively lower sensitivity[8,10,24]. Here, we reported a comprehensive survey on the phenotype and dynamics of diverse ISC subsets in different phases after high-dose irradiation. Generally, we identified three subpopulations of ISCs, including Lgr5+ CBCs, +4 RSCs as well as a cluster of Clu+ revSCs reported recently[11], with distinct dynamic characteristics. Additionally, our data provides new insights into the interconversion relationships among the three ISC subpopulations. Although a previous study has shown Lgr5+ CBCs could give rise to +4 RSCs in culture[10], our data suggests this conversion could be induced in vivo by intestinal injury. Besides, we also showed that Clu+ revSCs may originate from Lgr5+ CBCs and +4 RSCs, although further studies are still needed to test and verify the transformation processes.

Exposure to irradiation could also strongly affect immune system responses in the intestine, which were essential for maintaining mucosal homeostasis. In this study, we observed that Cx3cr1+ resident macrophages were drastically reduced at 1 day post irradiation and were then followed by a massive influx of monocytes and macrophages into the injured intestine. Previous report suggested that the macrophage pool in the intestine of adult mice requires constant replenishment from circulating monocytes[14] and resident and inflammatory macrophages would represent alternative differentiation outcomes of the same precursor[34,53]. Here our data revealed the decrease of resident population and increase of inflammatory population in the radiation-induced injury and these results suggested that the bidirectional differentiation of peripheral Ly6c+ monocytes to pro-inflammatory macrophages or Cx3cr1+ resident macrophages may be important for balancing the radiation-induced inflammation and resident macrophages recovery. We also showed that different TFs may contribute to the alternative fates of Ly6c+ monocytes, which could be potential intervention targets for modulating the early stage inflammation induced by radiation.

In addition, we found the cellular interacting patterns varied greatly at different times during RIII and recovery; especially, the massive interactions between the macrophages and endothelial cells at early inflammation stage aroused our attention. Our results showed that the cross-talk between pro-inflammatory macrophages and activated endothelial cells could achieve a quick amplification of inflammation upon RIII. Endothelial cells have been reported to play a crucial role in radiation-induced gastrointestinal syndrome. In the context of whole-body irradiation models, microvascular endothelial apoptosis was considered the primary pathological event leading to stem cell dysfunction when exposed to lethal doses[54]. To gain a deeper understanding of the characteristics of the endothelial cell-macrophage axis under such circumstances, further research endeavors are imperative.

As the primary component of intraepithelial lymphocytes, T cells play a fundamental role in maintaining dynamic immune homeostasis of the gastrointestinal tract. Dysregulated T cells are considered major pathogenic cells in IBD[15]. Here our result further demonstrated the increase of TEs in the early inflammation stage of RIII. Through CCC analysis, several key signaling pathways involved in the intercellular interactions between T cells and other intestinal cell types were identified, among which the CD200 signaling pathway was specifically associated to the CD8+ TE cells with endothelial and stromal cells. CD200-CD200R interaction has been reported to initiate an inhibitory pathway that result in immunosuppressive and anti-inflammatory effects[55]. Further in vivo and in vitro mechanism experiments are required to confirm its regulation effect on inflammation development in the process of RIII.

Due to the constrained availability of tissue samples and ethical reasons, it's rather difficult to directly study RIII in humans. Animal models, especially mouse models, have played a critical role in our understanding of the disease to date. The gastrointestinal tracts in both human and mouse are composed of organs which are anatomically and developmentally similar, expect for the specific differences in structure and morphology[56,57]. For comparison of cell types in human and mouse intestine, we explored the scRNA-seq expression profiles of human intestine[17] and found that subtypes of stem cells and macrophages in humans showed high similarity with those found in mice (Supplementary Fig. 8). Besides, several key regulators identified in our data have already been demonstrated to be pivotal for cell functions in humans, including HMGA1 and RUNX1, which were reported to regulate the differentiation in human stem cells[58,59]; and STAT1 and IRF1, which were observed to participate in the cytokine production in human inflammatory macrophages[60,61]. We expect that our findings would help to provide insights into the RIII of human intestines.

In summary, based on scRNA-seq, our research refined the radiosensitivity of small intestinal cells in the cell subtype level.

Besides, our dataset revealed the dynamic patterns of three ISC subpopulations and their interconversion relationships. Additionally, we showed the bidirectional differentiation of peripheral monocytes into pro-inflammatory macrophages and resident macrophages, and the amplifying communicating relationships between macrophages and endothelial cells during the inflammatory response upon RIII. We also identified different subtypes of T cells and revealed the relevant immunomodulation pathways they would participate in during the early stage of RIII. These findings may provide new views on the cellular and molecular mechanisms of RIII.

## Methods

**Ethics statement**. The study was approved by the institutional review board of Beijing Institute of Radiation Medicine (protocol ID: IACUC-DWZX-2020-623). We have complied with all relevant ethical regulations for animal testing. The work submitted in this article was solely completed by the teams of Gangqiao Zhou, Yiming Lu and Yang Li, and it is original. Excerpts from others' work have been clearly identified and acknowledged within the text and listed in the list of references.

**Mice and groups**. Wild-type male C57BL/6 J mice (weight of $20 \pm 2$ g) were purchased from Beijing Vital River Laboratory Animal Technology Co., Ltd. (Beijing, China). All the mice were bred in a specific pathogen-free environment under conditions of constant temperature of $22 \pm 1°$ C, relative humidity of 60%, and regular dark-light schedule (lights on from 7 a.m. to 7 p.m.) at the Experimental Animal Center of the Beijing Institute of Radiation Medicine, China. A total of 97 mice were used in this study, including 78 random mice receiving abdominal irradiation and 19 random mice as negative controls. The irradiation exposed mice were then randomly divided into four groups and intestinal samples were collected and used for experiments at day 1, 3, 7, and 14 post irradiation, respectively. Specifically, 17 mice were used for scRNA-seq ($n = 3$ or 4 for each group), 20 mice for flow cytometry ($n = 3$ for each group, with additional 5 mice for flow cytometry cell sorting at day 3), 30 mice for immunohistochemistry ($n = 6$ for each group), and 30 mice for in situ hybridization analyses ($n = 6$ for each group).

**Abdominal irradiation of mice**. Mice were anaesthetized with an intraperitoneal injection of 0.5% pentobarbital (43 mg/kg body weight) and exposed to a single dose of 15 gray (Gy) abdominal irradiation (from the xiphoid process to the pubic symphysis with lead bricks to cover the other parts of the mice) using a $^{60}$Co irradiator to induce intestinal injury in vivo. All experiments were repeated at least twice with $n = 6$ mice in each group, except for the scRNA-seq, in which each group contain $n = 3$ or 4 mice. Mice were monitored for up to 15 days, and the changes in small intestine lengths and weights were recorded on day 0, 1, 3, 7, 14 post irradiation.

**Morphological analyses of mice villus and crypt**. Mice were monitored for up to 15 days. The mice in control and irradiation groups were sacrificed on day 1, 3, 7, 14 following irradiation, and the intestine tissues were harvested and fixed in a 10% neutral buffered formalin solution. The fixed samples were dehydrated, cleared and permeated with paraffin in a tissue processor, and subsequently embedded in paraffin blocks using an embedding system. Paraffin-embedded samples were sectioned at a 5-mm thickness and stained with hematoxylin & eosin (HE) staining. The slides were imaged at $50\times$, $100\times$ and $400\times$ magnification, respectively, using an Olympus BX51 microscope (Japan). The villus length and width, crypt depth, thickness of muscular layer

and number of crypts per intestinal length were measured with ImageJ (NIH). The villus length was measured from the top to the base of the villus at the entrance to the intestinal crypt. The villus width was measured at half of its length. The crypt depth was measured from the depth of the invagination to the adjacent villi. For each group, at least 30 well-oriented villi were measured and the mean value was calculated.

**Single-cell sequencing library construction**. To isolate single cells, duodenums were amputated (about 5 cm beyond the pylorus) and 10 cm jejunum segments following the incision of the mice on day 0, 1, 3, 7, 14 after irradiation were washed in cold phosphate buffered saline (PBS), cut longitudinally into roughly 2-mm-long pieces and were isolated using the Liver Dissociation Kit (mouse) and GentleMACS Dissociator (Miltenyi Biotec, Germany) according to the manufacturer's instructions. Tissues were filtered through a 40-μm-mesh cell strainer on ice, pelleted by centrifugation at 4 °C and washed twice with the ice-cold regular medium to remove the debris. An estimated 5000 single cells per sample were then subjected to 10x Genomics single-cell isolation and RNA sequencing following the manufacturer's recommendations. Illumina HiSeq 3000 was used for deep sequencing. Two technical replicates were generated per sorted cell suspension.

**Analysis of scRNA-seq data**. The Cell Ranger software (version 2.2.0) provided by 10x Genomics was used to align the reads from droplet-based scRNA-seq to an indexed mouse genome (mm10, NCBI Build 38), generating a digital gene expression matrix (UMI counts per gene per cell) for each sample. Expression matrices for all samples were filtered, normalized, integrated and clustered using the standard Seurat (version 2.3.4) package procedures. More exactly, for the first quality-control (QC) step, we took the following measures: (i) removal of cells with an expression level of <500 or >8000 genes; (ii) filtering out cells with >10% of reads mapping to mitochondrial genes; (iii) eliminating genes detected in <3 cells. After applying these QC criteria, a total of 22,680 cells and 19,588 genes in total remained and were included in the following analyses. Then the expression matrices were log-normalized and scaled to remove the unwanted variation from the total cellular read count and mitochondrial read count, as implemented in Seurat's *NormalizeData* and *ScaleData* functions. To integrate datasets from different samples, we used a subset of highly variably expressed genes to perform the CCA. First, the top 1000 genes with the largest dispersion in each dataset were selected; then the genes from all 17 datasets were intersected to determine an overlap-gene set. We used the overlap-gene set as variable genes to implement CCA through Seurat's *RunMultiCCA* function, which returned an integrated Seurat objects with canonical correlation vectors. Seurat's *AlignSubspace* function was employed to align the top ten dimensions in CCA subspaces and generate the *cca.aligned* dimensional reduction. With the first ten components of the dimensional reduction, we then performed a SNN modularity optimization clustering method with the Louvain algorithm as implemented in the *FindClusters* function, which finally identified 54 clusters of different cell types or subtypes, and different clusters of cells were visualized using a further tSNE dimensionality reduction. In parallel, we used Scrublet[62] (version 0.2.3) to investigated the possibility of doublets in the dataset. Doublet scores for single cells were determined for each sample individually, with the expected doublet rate at 0.06 and the thresholds were adjusted after visual inspection of doublet score distributions.

To test the robustness and reliability of the annotation result, we re-analyzed the data with the latest version of Cell Ranger

(version 7.1.0) and the updated Seurat packages (version 4.3.0). Gene expression matrix for the same cells were generated with the *cellranger count* function and further processed with the Seurat v4 pipeline. The expression matrices were normalized separately and further merged into an integrated object. Variable features were selected with the *SelectIntegrationFeatures* function and then scaled to remove unwanted biases. Principal components analysis (PCA) dimension reduction was conducted, on which tSNE dimension reduction and SNN-based clustering were performed.

**Immunohistochemistry assay.** The 5 μm thick sections from the paraffin-embedded small intestine sections were deparaffinized and rehydrated using xylene and ethanol and boiled for 15 min (min) in 10 mM citrate buffer solution (pH 6.0) for antigen retrieval. The sections were then immersed in a 3% hydrogen peroxide solution for 10 min to block the endogenous peroxidase. Slides were incubated with goat serum for 10 min and then with the primary antibody anti-Ki67 (#ab16667, 1:200; Abcam). A horse radish peroxidase (HRP)-based signal amplification system was then hybridized to a goat anti-rabbit IgG (H&L) secondary antibody (#PV9001; ZSBIO) followed by colorimetric development with diaminobenzidine (DAB). Positive cells were counted in the crypt and villi at 30 randomly selected position per group with ImageJ.

**TUNEL assay.** Apoptotic cells were identified by terminal deoxynucleotidyl transferase-mediated dUTP nick end-labeling (TUNEL) staining using the In Situ Cell Death Detection Kit (Roche) according to the manufacturer's protocol. Briefly, the paraffin-embedded sections were prepared the same as hematoxylin & eosin (HE) staining. After drying and deparaffinized, the section was treated by proteinase K (20 μg/mL; Roche, Swiss) for l0 min to make the cell membrane permeable, and then treated by the mixed reaction solution for TUNEL reaction. After treatment with biotin-labeled HRP, DAB chromogen was used to render the color. TUNEL-positive cells were identified, and their numbers were counted within a defined area (μ2) using an Olympus BX51 microscope (Japan).

**In situ hybridization.** In situ hybridization (ISH) assays for *Olfm4* were performed with the RNAscope kit (Advanced Cell Diagnostics, California, USA) according to the manufacturer's instructions. Five μm formalin-fixed, paraffin embedded tissue sections or 8 μm OCT frozen were pretreated with heat and protease digestion prior to hybridization with target probe,s (Advanced Cell Diagnostics). An HRP-based signal amplification system was then hybridized to the target probes followed by colorimetric development with DAB or Fitc, and Cy3. The housekeeping gene *ubiquitin C* (*UBC*) was served as a positive control and the *dapB* gene, which is derived from a bacterial gene sequence, was used as a negative control.

**Identification of marker genes.** To identify the marker genes for each of the 54 clusters of different cell types or subtypes, we contrasted the cells from each cluster to cells from all the other clusters using the Seurat's *FindMarkers* function. The marker genes were required to: (1) have an averaged expression in the current cluster that is at least 0.25-fold (log-scale) larger than that in all other clusters; and (2) show expression in at least 10% of cells in either of the two comparative populations.

**Flow cytometry.** Single cells prepared as above were resuspended in PBS. For macrophages, the antibodies used were against Cd45, F4/80, Cd11b, and Cd14, and cells were stained with 7-AAD to differentiate live cells. Antibody staining was performed at RT for 30 min before washing cells twice with PBS. The 7-AAD was added to the final FACS medium for 10 min before flow cytometry analyses and/or isolation of required single-cell populations. These cells were gated on live cells (those that were 7-AAD negative) and CD45 positivity. Subsequent determinations included the presence of F4/80 and Cd11b followed by the presence or absence of Cd14. The cells were analyzed on a BD FACSCanto II flow cytometer (BD, New Jersey, USA) utilizing FACS Diva Software (BD).

**RNA velocity analyses.** RNA velocity of single cells was estimated as previously reported[25]. More exactly, the spliced and unspliced transcript reads were calculated based on the Cell Ranger output using the velocyto command line tool with run10x subcommand. Transcript counts from different samples were merged and genes with an average expression magnitude <0.5 (for spliced transcripts) or <0.05 (for unspliced transcripts) in at least one of the clusters of each cell type were then removed. Cell-to-cell distance was calculated using Euclidean distance based on the correlation matrix of the *cca.aligned* dimensional reduction from Seurat. RNA velocity was estimated using gene-relative models, with k-nearest neighbor cells pooling of 20 and fit quantile of 0.02. Velocity fields were then visualized in the tSNE dimensionality reduction from Seurat.

**Gene set variation analysis (GSVA).** To assess the relative pathway activity on the level of individual cells, we conducted the gene set variation analysis (GSVA) with the GSVA package (version 1.32.0)[63]. The 50 hallmark gene sets (version 6.2) representing specific well-defined biological states or processes were obtained from the Molecular Signatures Database (MSigDB)[64], and genes in each set were converted to orthologous genes in mouse with g:Profiler[65]. GSVA was performed in the *gsva* function with standard settings. We then fit a linear model to the output gene set-by-cell pathway enrichment matrix, implemented in the *lmFit* function of limma package (version 3.40.6), to detect the differentially enriched pathways.

**SCENIC analysis.** The pySCENIC (version 0.9.11)[26] analysis was run as described for the stem cells and myeloid cells. The normalized expression matrix exported from Seurat was set as the input of pySCENIC pipeline. In addition, the genes with less than 200 UMI counts or detected in less than 1% of the cells in the corresponding cell types were removed as noise. Two TF ranking databases, namely the *TSS ± 10 kb* and *500bpUp100Dw* databases, were used for inference of co-expression modules and identification of direct targets. A regulon-by-cell matrix was then calculated to measure the enrichment of each regulon as the area under the recovery curve (AUC) of genes defining the regulon. To identify the key regulons driving the differentiation trajectories among the stem or myeloid cell subpopulations, cells along each RNA velocity flow were selected and their tSNE coordinates were converted using PCA to capture the differentiation process on the first principal component (PC1). Differential enrichment was calculated on the SCENIC AUC matrix along the PC1 coordinates for each trajectory with a general additive model implemented in the gam package (version 1.16.1).

**Cell-cell communication analysis.** We used CellChat (Version 1.6.1)[66] to infer the CCC among different cell types following the tutorials in the CellChat github repository. In brief, a CellChat object was constructed based on the normalized expression profile at each time point post irradiation. Cellular communication probability was then computed with the default parameters and CCC with <10 cells in the corresponding cell groups was filtered

out. Communication probability on signaling pathway level was further inferred by summarizing the communication probabilities of all ligands-receptors interactions associated with each signaling pathway.

**Statistics and reproducibility**. GraphPad Prism 5 and R (version 3.6.0) were used to perform the statistical analyses. The Kaplan–Meier method was used to analyze the animal survival curves. The Chi-square test was applied to analysis the distribution of cells at different times before and after irradiation. $P$-values of multiple testing were corrected by Bonferroni correction method and an adjusted $P$-value $< 0.05$ was considered to be statistically significant. $n = 4$ mice in the control group and $n = 4$, 3, 3, 3 mice in day 1, 3, 7, 14 after irradiation.

**Reporting summary**. Further information on research design is available in the Nature Portfolio Reporting Summary linked to this article.

## Data availability

The scRNA-seq dataset generated in this study are available at the National Center of Biotechnology Information's Gene Expression Omniobus database under the following accession number: GSE165318. Source data underlying Figs. 1b, c, 2a, 3c, d, 4c, e, f, 5a, f, 6a, f are presented in Supplementary Data 1. Other data are available upon reasonable request to the corresponding authors

## Code availability

The R scripts used for analysis and visualization are available upon reasonable request to the corresponding author.

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

## Acknowledgements

This work was funded by the General Program (32270714, 81573251 and 81672369) of the Natural Science Foundation of China (www.nsfc.gov.cn), Beijing Nova Program (20180059), National Key R&D Program of China (No. 2017YFA0504301), Chinese Key Project for Infectious Diseases (No. 2018ZX10732202 and 2017ZX10203205) and Beijing Institute of Radiation Medicine (BIRM) Innovation Fund (BIOX0201).

## Author contributions

Y.L., G.Z., and Y.L. was the principal investigators who conceived and designed the study, obtained financial supports and approved the final version of the manuscript; H.L., H.Y., S.J., L.M., and J.P. performed the data analyses; H.Y. and X.L. conducted most of the cell sorting and functional experiments; H.Y., X.L., Y.X., Y.Y., H.Z., Y.H., and C.Y. performed abdominal irradiation of mice and collected intestinal samples; Y.L, G.Z., and Y.L. drafted the manuscript. All the authors read and approved the final version of the manuscript.

## Competing interests

The authors declare no competing interests.
