## [Peer Review File · Communications Biology]

Reviewers' comments:

Reviewer #1 (Remarks to the Author):

Communications Biology

Intestine is a complex organ consisting multiple cell type. Author has examined the radiation sensitivity of these multiple cell types using SS RNA seq in a time dependent manner. This manuscript provides some key information and should be helpful to identify targets to minimize radiation toxicity. However there are some concerns about radiation model and some experimental strategy.

Major comments:

Authors have used abdominal irradiation model using 15Gy single fraction. Abdominal irradiation model is clinically relevant. However, it is important to use fractionated irradiation as all SBRTs are hypo or hyper fractionated for abdominal radio therapy.

Whole body irradiations are always considered more appropriate to study Radiation induced gastrointestinal syndrome. This will also allow to study involvement of bone marrow in GI syndrome. To address GI syndrome from radiation accident or terrorism whole body irradiation model is more appropriate.

ssRNA seq data showed the differences in radiosensitivity among +4 , CBCs and +Clu differentiated pool. This will be very informative to determine their presence in supralethal (LD100/10), lethal (LD100/20) dose level. It is critical to identify which ISC population is absolutely indispensable for recurrence of stem cell regeneration. Immuno histochemistry should be sufficient.

Author have mentioned that Cx3CR1 positive cells are more radiosensitive than Ly6C+ cells as later one is more abundant in intestine post abdominal irradiation. In response to abdominal irradiation Ly6C+ inflammatory monocytes are recruited from circulation. Arguably these cells were not exposed to irradiation. To compare the radiosensitivity between Ly6C+ and Cx3CR1 cells it is important to consider whole body irradiation model.

Endothelial cells are always part of inflammatory process. However, previous report suggested that endothelial cells are highly radiosensitive and critical for epithelial recovery against lethal doses (Paris F, Fuks Z, Kang A, Capodiceci P, Juan G, Ehleiter D, Haimovitz-Friedman A, Cordon-Cardo C, Kolesnick R. Endothelial apoptosis as the primary lesion initiating intestinal radiation damage in mice. *Science*. 2001;293:293-297.). Therefore, it is important to determine endothelial cell macrophage axis in response to lethal doses (LD100/10).

Analysis and discussion on T cell subtypes and their radiosensitivity is needed.

Reviewer #2 (Remarks to the Author):

This paper describes the kinetics of intestinal tissue recovery after high dose of radiation by single-cell RNA-Seq. In this manuscript, the authors used RNA velocity to reveal the cellular dynamics of tissue stem cells and immunocompetent cells in the microenvironment. Radiation effects and the microenvironment induced by high-dose irradiation, especially on immune response and inflammation, have been considered important issues, however; the details mechanisms have not been clarified so far. The results of this study are important for understanding of interconversion of

cells induced by irradiation in intestines because they provide insight into the time scale of immune and inflammatory responses. On the other hand, there are some questions about the description and interpretation of the data, so the manuscript needs to be revised.

Major comments:

The term "multicellular ecosystem" is confusing; an ecosystem is usually thought of as a collection of diverse species. The term ecosystem in the gut is generally reminiscent of the gut microbiota, and should be reconsidered to make it clear that the focus is on cells derived from mouse tissue, such as "multicellular relationship" or "multicellular dynamics" and so on.

The authors classified stem cells into five fractions based on scRNA-Seq results. The direction of cellular dynamics was indicated by RNA velocity. In the stem cell fraction, +4 RSCs and revSCs showed unique clusters by its gene expression. On the other hand, the authors classified CBC clusters into three clusters. Seemingly, only one of them (c4) suggests interconversion with +4 RSCs. However, the other two (c9 and c16) are not associated with +4 RSCs. I also doubted why the authors did not show the Lgr5 gene to classify as a cluster in CBC. I think it is difficult to argue that there is a direction from CBC to +4 RSC on the suggested this evidence.

Figure 1

Fig1C: Vertical axis should be "Body" weight so that we are able to know what the weight is.

Fig.1E: Cell Type is listed on the right side with the figure distinguished by color, however; text should also be written at the top of the figure, as in Fig.2A for better understandings.

Fig.1F/1G: The information of all the cell categories was summarized in a pie chart, but since the composition ratio differs between non-irradiated and irradiated cells, the information should be presented separately such as "Day 0" and "Day 1". On the other hand, as shown in Fig. S1, cell numbers decrease significantly irradiation. Therefore, the composition ratio should be different when normalized by the number of cells at each time point. Is it possible to show the composition ratio considering the total number of cells after irradiation?

Minor comments:

Line 107: The description of radioisotope should be as ⁶⁰Co or Co-60, not be Co⁶⁰.

Line 115: S1D -> S1E

Line 117: S1E -> S1F

Line 119: S1F -> S1G

Reviewer #3 (Remarks to the Author):

This study presents a comprehensive dynamic landscape of the cellular microenvironment during intestinal injury and regeneration by combining single-cell transcriptome profiling with temporal distribution analysis, lineage reconstruction, TF profiling, and cellular interaction analyses. The researchers generated transcriptomes of 22,680 single cells in the intestinal microenvironment to profile the dynamics of ISCs and immune cells located in the mucous layer. The results show a dynamic single-cell map of the multicellular ecosystems in the healthy and injured small intestines. The data can be a valuable resource for further investigation of the cellular mechanism of radiation-induced intestinal injury and the development of potential therapeutic strategies.

The work presented in this study is convincing, as it is based on a large dataset generated through single-cell transcriptome profiling and other analyses. However, further evidence would be required to strengthen the conclusions, such as validation of potential therapeutic targets through in vivo experiments.

As of the current writing date, cellranger has been updated to version 7.x, which includes a significant number of fundamental updates. However, the authors used an outdated version of cellranger, specifically version 2.2.0. Similar situation for Seurat as well.

The QC step of scRNAseq data analysis should also be more rigorous. For example, include metrics about mitochondrial genes.

It will be also great to include a discussion of how well the findings from this study may translate to humans, as it was conducted on mice.

Revision of COMMSBIO-23-1149

Title: Single-Cell Map of Dynamic Cellular Microenvironment of Radiation-Induced Intestinal Injury

Point by point reply to the reviewers.

We'd like to thank the reviewers for your helpful comments. We greatly appreciate the input, and we have enclosed a revised version that will hopefully address the concerns raised by the reviewers. Particularly, we conducted additional experiments to determine the lethal dose and to detect the reaction of three ISC subtypes, performed more in-depth analysis of T cell subtypes to provide a comprehensive understanding of their characteristics and functions, contrasted the cluster results obtained from both older and the latest versions of the analysis tools to ensure the robustness of results, and compared the characteristics of cell types between human and mouse intestines to demonstrate the relevance and applicability of our findings. We detail our responses to the reviewers below.

In response to Reviewer #1, we greatly appreciate your time and input and are grateful for the very constructive criticism. In response to your specific comments:

“Intestine is a complex organ consisting multiple cell type. Author has examined the radiation sensitivity of these multiple cell types using SS RNA seq in a time dependent manner. This manuscript provides some key information and should be helpful to identify targets to minimize radiation toxicity. However there are some concerns about radiation model and some experimental strategy.”

Major comments:

1. “Authors have used abdominal irradiation model using 15Gy single fraction. Abdominal irradiation model is clinically relevant. However, it is important to use fractionated irradiation as all SBRTs are hypo or hyper fractionated for abdominal radio therapy.”

Response: We thank the reviewer for your valuable suggestion. While clinical abdominal radiotherapy is typically fractionated, it's noteworthy that the use of a single abdominal irradiation fraction has also been a well-accepted and established model for studying radiation-induced gastrointestinal injuries in the research community. This approach, particularly in mouse models, has been extensively utilized to investigate various aspects of gastrointestinal injuries and treatment strategies for tumors. For instance, it has been used to investigate mitigation and therapeutic strategies for addressing radiation enteritis (PMID: 37326937) and gastrointestinal toxicity (PMID: 37531370) induced by gamma radiation. Besides, a study focusing on postoperative endometrial carcinoma has demonstrated that, compared to schedules involving 2 or 3 fractions of lower dose, a single dose of 7 Gy is both safe and effective, which offers similar outcomes in terms of vaginal-cuff relapses, complications, and patient comfort, while reducing hospital attendance (PMID: 31865604). Moreover, fractionated irradiation also has its own issues. For example, the cumulative effect of radiation exposure may lead to chronic tissue damage, inflammation, and changes in the tissue microenvironment (PMID: 29551910). Besides, frequent handling and repeated radiation exposures may cause chronic stress to the mice, potentially affecting their physiology and response to treatment. In light of these factors, we made the choice to employ a single-fraction abdominal irradiation model with a dose of 15 Gy. This approach allows us to gain valuable insights into the

acute phase of intestinal injury and repair while minimizing potential confounding factors associated with fractionated regimens.

2. “Whole body irradiations are always considered more appropriate to study Radiation induced gastrointestinal syndrome. This will also allow to study involvement of bone marrow in GI syndrome. To address GI syndrome from radiation accident or terrorism whole body irradiation model is more appropriate.”

Response: We greatly appreciate your valuable input. We totally agree with the reviewer that whole body irradiations would be more suitable for exploring the involvement of bone marrow in gastrointestinal syndrome. Indeed, unexpected nuclear accidents or acts of terrorism can lead to multi-organ irradiation scenarios. Our study primarily focuses on understanding the effects of abdominal irradiation. We apologize for any inappropriate descriptions regarding unexpected radiation accidents in the original manuscript and have accordingly modified the context in the revised version.

3. “ssRNA seq data showed the differences in radiosensitivity among +4, CBCs and +Clu differentiated pool. This will be very informative to determine their presence in supralethal (LD100/10), lethal (LD100/20) dose level. It is critical to identify which ISC population is absolutely indispensable for recurrence of stem cell regeneration. Immunohistochemistry should be sufficient.”

Response: It's a good point to examine the presence of different ISC subtypes in supralethal (LD100/10) and lethal (LD100/20) dose levels. To this end, we conducted additional experiments to determine the supralethal and lethal doses and detected the reaction of three ISC subtypes under these doses. Ultimately, after testing a variety of doses, we confirmed a dose of 20 Gy as the supralethal dose (Figure R1A). We observed a 100% mortality rate within 6 days. Therefore, the sampling period post-modeling extended only up to 3 days. We preferentially selected qPCR to detect the markers of three types of stem cells: *Lgr5* for crypt-based columnar cells (CBCs), *Hopx* for reserve stem cells, and *Clu* for revival stem cells, to quantify the abundances of three stem cells (Figure R1B), combined with immunohistochemical staining of *Olfm4* for CBCs to study their radiosensitivity (Figure R1C). Our findings indicate that CBCs were the most radiosensitive stem cells in intestine crypts. It is noteworthy that the lethal dose in abdominal irradiation differs from that in whole body irradiation. In the abdominal irradiation model, radiation enteritis-induced fatalities typically occur in the early stage (≤ 7 days post irradiation), after which the damaged intestine tends to initiate a repair process. Given this context, we did not establish a clear demarcation between supralethal (LD100/10) and lethal (LD100/20) dose levels during the preliminary experiments.

Figure R1. (A) Survival rate of mice following abdominal irradiation with 20 Gy. (B) *Lgr5*, *Hoxp* and *Clu* mRNA expression in jejunum. (C) Representative microscopic images of jejunal sections immunostained with anti-Olfm4 antibody to assess Olfm4 expression.

4. “Author have mentioned that Cx3CR1 positive cells are more radiosensitive than Ly6C+ cells as later one is more abundant in intestine post abdominal irradiation. In response to abdominal irradiation Ly6C+ inflammatory monocytes are recruited from circulation. Arguably these cells were not exposed to irradiation. To compare the radiosensitivity between Ly6C+ and Cx3CR1 cells it is important to consider whole body irradiation model.”

Response: This is an excellent question. We concur with the reviewer's recommendation that employing a whole-body irradiation model is essential to assess the radiosensitivity disparity between Ly6C+ and Cx3cr1 cells. Our present data would suggest that Cx3cr1 cells might be more radiosensitive than Ly6C+ cells. However, further validation of this argument requires experiments involving whole-body irradiation. We have modified the statement in the **Result** section (page 14, lines 286–289) in the revised manuscript accordingly.

5. “Endothelial cells are always part of inflammatory process. However, previous report suggested that endothelial cells are highly radiosensitive and critical for epithelial recovery against lethal doses (Paris F, Fuks Z, Kang A, Capodiec P, Juan G, Ehleiter D, Haimovitz-Friedman A, Cordon-Cardo C, Kolesnick R. Endothelial apoptosis as the primary lesion initiating intestinal radiation damage in mice. *Science*. 2001;293:293-297.). Therefore, it is important to determine endothelial cell macrophage axis in response to lethal doses (LD100/10).”

Response: We thank the reviewer for your excellent guidance. We recognize the importance of exploring the endothelial cell-macrophage axis in response to lethal doses and we are grateful for your insight which provides us with a valuable avenue for more in-depth study. As reported in the reference the reviewer mentioned, endothelial cells play a crucial role in radiation-induced gastrointestinal syndrome. In the context of whole-body irradiation models, microvascular endothelial apoptosis is considered the primary pathological event leading to stem cell dysfunction (*Science*. 2001; 293: 293–297). In our study, we observed a novel role of endothelial cells in the abdominal irradiation model, where they actively contribute to the amplification of early inflammatory response through the recruitment of macrophages. As suggested by the reviewer, we are committed to exploring the characteristics of the endothelial cell-macrophage axis in response to lethal doses in our future research endeavors. We are dedicated to conducting comprehensive

studies that will address this question thoroughly. These discussions have been clarified in the **Discussion** section (page 20, lines 430–435) in the revised manuscript.

6. “Analysis and discussion on T cell subtypes and their radiosensitivity is needed.”

Response: We thank the reviewer for your constructive suggestions. As suggested by the reviewer, we have conducted more in-depth analyses of T cell subtypes, including the phenotype and functional annotation to identify different subtypes, exploration of their dynamics to investigate the radiosensitivity, and examination of cell-cell communication to discern their potential roles in the early inflammation stage. Briefly, we identified Cd4+ naïve T cells (TNs), Cd8+ effector (TEs) and tissue-resident memory T cells (TRMs), and proliferative T cells. Notably, Cd8+ TRMs were the predominant T cell subtype in the control group but exhibited a significant reduction from days 1 to 7 post-irradiation, with restoration observed by day 14. In contrast, Cd8+ TEs and Cd4+ TNs demonstrated a remarkable increase on day 3 post-irradiation. Intercellular communication analysis revealed that these T cell subtypes would participate in the inflammation regulation through multiple signaling pathways associated with immune responses. Description and discussion about these new results have been added to the **Result** (page 17, lines 355–384) and **Discussion** (page 21, lines 436–447) sections in the revised manuscript.

In response to Reviewer #2, we would like to thank the reviewer for taking your time to evaluate our work. We are grateful for the positive feedback and addressed the issues as follows:

“This paper describes the kinetics of intestinal tissue recovery after high dose of radiation by single-cell RNA-Seq. In this manuscript, the authors used RNA velocity to reveal the cellular dynamics of tissue stem cells and immunocompetent cells in the microenvironment. Radiation effects and the microenvironment induced by high-dose irradiation, especially on immune response and inflammation, have been considered important issues, however; the details mechanisms have not been clarified so far. The results of this study are important for understanding of interconversion of cells induced by irradiation in intestines because they provide insight into the time scale of immune and inflammatory responses. On the other hand, there are some questions about the description and interpretation of the data, so the manuscript needs to be revised.”

Major comments:

1. “The term “multicellular ecosystem” is confusing; an ecosystem is usually thought of as a collection of diverse species. The term ecosystem in the gut is generally reminiscent of the gut microbiota and should be reconsider to make it clear that the focus is on cells derived from mouse tissue, such as “multicellular relationship” or “multicellular dynamics” and so on.”

Response: We thank the reviewer for your great suggestions. In the revised manuscript we have modified the term “multicellular ecosystem” to “cellular microenvironment”, “multicellular relationship”, or “multicellular dynamics” contextually.

2. “The authors classified stem cells into five fractions based on scRNA-Seq results. The direction of cellular dynamics was indicated by RNA velocity. In the stem cell fraction, +4 RSCs and revSCs showed unique clusters by its gene expression. On the other hand, the authors classified CBC clusters into three clusters. Seemingly, only one of them (c4) suggests interconversion with +4 RSCs.

However, the other two (c9 and c16) are not associated with +4 RSCs. I also doubted why the authors did not show the *Lgr5* gene to classify as a cluster in CBC. I think it is difficult to argue that there is a direction from CBC to +4 RSC on the suggested this evidence.”

Response: We understand the reviewer’s concern about interconversion among the ISC subtypes and the expression of *Lgr5*, the canonical marker of CBCs, across stem cell subpopulations. Due to the sequencing bias of scRNA-seq for gene expression, the detection sensitivity of *Lgr5* mRNAs is relatively low, consistent with previous scRNA-seq studies (PMID: 35479413, 35176508). In our analysis, we observed high expression levels of other *Lgr5*+ CBC markers, including *Olfm4*, *Smoc2*, and *Prom1*, in clusters C4, C9, and C16, leading us to refer to them as *Lgr5*+ CBCs. As shown in Figure 3B and 3C, there’s still subtle difference in the expression of marker genes and temporal distributions among the three CBC clusters, hinting at the existence of even more subtle heterogeneity among those CBCs. Notably, cluster C4 was positioned in close proximity to the +4 RSCs and Clu+ revSCs in the *t*SNE projection plot, indicating a closer relationship between C4 and the other two subtypes. While these observations are intriguing, we acknowledge that further *in vivo* and *in vitro* mechanistic experiments are necessary to rigorously test and validate the specific interconversion processes. In the revised manuscript, we have made appropriate modifications to the descriptions in the **Results** section (page 10, lines 198-203 and page 12 , lines 241-244) to reflect this acknowledgment of the need for additional experimental validation.

3. “Fig1C: Vertical axis should be “Body” weight so that we are able to know what the weight is.”

Response: We thank the reviewer for the kind suggestions and have modified Fig.1C as suggested.

4. “Fig.1E: Cell Type is listed on the right side with the figure distinguished by color, however; text should also be written at the top of the figure, as in Fig.2A for better understandings.”

Response: We thank the reviewer for the kind suggestions and have added cell type label on the top of Fig.1E as suggested.

5. “Fig.1F/1G: The information of all the cell categories was summarized in a pie chart, but since the composition ratio differs between non-irradiated and irradiated cells, the information should be presented separately such as “Day 0” and “Day 1”. On the other hand, as shown in Fig. S1, cell numbers decrease significantly irradiation. Therefore, the composition ratio should be different when normalized by the number of cells at each time point. Is it possible to show the composition ratio considering the total number of cells after irradiation?”

Response: It’s a good point to explore the composition ratio separately and normalize it by the number of cells at each time point. As suggested by the reviewer, we have updated the manuscript by providing a more detailed and clear representation of the changes of cell composition ratios over the five time points. Specifically, we have included separate pie charts for each time point in the revised Supplementary Figure S4A, illustrating the composition of cell types with greater detail. Additionally, we have introduced a stacked bar plot in the revised Supplementary Figure S4B, which presents the fraction of cell types at each time point while normalizing it to the total number of cells before and after irradiation. We expected this modification will enhance the clarity of the presentation, making it easier to observe changes in the composition ratios across the five time points.

Minor comments:

1. "Line 107: The description of radioisotope should be as ^{60}Co or Co-60, not be Co^{60} ."
2. "Line 115: SID -> S1E"
3. "Line 117: S1E -> S1F"
4. "Line 119: S1F -> S1G"

Response: We thank the reviewer for the careful inspection and have corrected the corresponding descriptions in the revised manuscript.

In response to Reviewer #3, we greatly appreciate your time and input. In response to your specific comments:

"This study presents a comprehensive dynamic landscape of the cellular microenvironment during intestinal injury and regeneration by combining single-cell transcriptome profiling with temporal distribution analysis, lineage reconstruction, TF profiling, and cellular interaction analyses. The researchers generated transcriptomes of 22,680 single cells in the intestinal microenvironment to profile the dynamics of ISCs and immune cells located in the mucous layer. The results show a dynamic single-cell map of the multicellular ecosystems in the healthy and injured small intestines. The data can be a valuable resource for further investigation of the cellular mechanism of radiation-induced intestinal injury and the development of potential therapeutic strategies."

1. "The work presented in this study is convincing, as it is based on a large dataset generated through single-cell transcriptome profiling and other analyses. However, further evidence would be required to strengthen the conclusions, such as validation of potential therapeutic targets through in vivo experiments."

Response: We express our gratitude to the reviewer for your valuable and constructive suggestions. Due to the constrained availability of tissue samples and ethical reasons, it's rather difficult to directly study RIII in humans. Animal models, especially mouse models, have played a critical role in our understanding of the disease and investigation of the potential therapeutic targets. Although we did not directly perform in vivo experiments, we found that several key regulators identified in our data have already been demonstrated to be pivotal for cell functions in humans. For example, HMGA1 and RUNX1 have been reported to regulate the differentiation in human stem cells (PMID: 23166588, 25894653); and STAT1 and IRF1 have been observed to participate in the cytokine production in human inflammatory macrophages (PMID: 32661180, 24012417). These findings strongly imply that these regulators hold promise as potential targets for intervention in RIII. Our forthcoming research efforts will be dedicated to conducting mechanistic experiments to validate their key roles. These discussion have been added to the **Discussion** (page 21, lines 448–461) section in the revised manuscript.

2. "As of the current writing date, cellranger has been updated to version 7.x, which includes a significant number of fundamental updates. However, the authors used an outdated version of cellranger, specifically version 2.2.0. Similar situation for Seurat as well."

Response: This is an excellent question. Tools and algorithms for analysis of scRNA-seq data have always been developing rapidly and fundamental updates are implemented in the new versions. To

test the robustness and reliability of our result, in the revised manuscript we have analyzed the scRNA-seq data with the updated version of Cell Ranger (version 7.1.0, the newest version available from 10X genomics) and Seurat (version 4.3.0, the stable version available from CRAN) and compared the new result with the original analysis. We found that the cell types identified using the updated versions closely aligned with those identified using the previous versions, as demonstrated in the revised Supplementary Figure S3. For the specific cell types we focused on in the paper, including stem, myeloid, and T cells, the subtypes identified with the current versions were well detected in the result from the updated versions, as shown in the following Figure R2. We believe that this additional analysis provides further confidence in the robustness and reliability of our results. We are grateful to the reviewer for raising this important point, and we hope this information addresses your concerns effectively.

Figure R2. (A-C) Heatmap of the average expression of the selected ISC (A), myeloid (B), and T cells (C) function-related marker genes in the relevant clusters identified with the updated versions of analysis tools.

3. “The QC step of scRNAseq data analysis should also be more rigorous. For example, include metrics about mitochondrial genes.”

Response: We totally agree with the reviewer’s view on the importance of the QC step in scRNA-seq data analysis. We had included in the **Materials and Methods** section a perhaps too brief description of QC steps we take when analyzing the data and we deeply apologize for this. We have now detailed the QC steps based on the metrics of total number of expressed genes, UMIs from the mitochondrial genome, and total number of detected cells. Specifically, we have taken the following measures to enhance data quality: (i) removal of cells with an expression level of less than 500 or greater than 8,000 genes; (ii) filtering out cells with more than 10% of reads mapping to mitochondrial genes; (iii) eliminating genes detected in less than three cells. Furthermore, we have thoroughly investigated the possibility of doublets and are pleased to report that the doublet rates in all clusters were remarkably low, remaining below 2% (as illustrated in the newly added Supplementary Figure S2D). We have updated and expanded the corresponding descriptions in the revised version (page 25, lines 536-539 in the **Materials and Methods** section and page 7, lines

140-142 in the **Results** section) to ensure better clarity and comprehension.

4. “It will be also great to include a discussion of how well the findings from this study may translate to humans, as it was conducted on mice.”

Response: This is a great suggestion. The gastrointestinal tracts in both human and mouse are composed of organs which are anatomically and developmentally similar, except for the specific differences in structure and morphology (PMID: 25561744, 31242501). To compare the cell types in human and mouse, we downloaded the scRNA-seq data for human intestine from Fawcner-Corbett, D. et al (PMID: 33406409), and explored the expression profiles of stem cells and macrophages. We found that subtypes of stem cells and macrophages in humans showed high similarity with those found in mice, as shown in the following Figure R3. There is limited research which utilized scRNA-seq to deconvolute the heterogeneity of human intestine, and the annotation resolution of intestinal cell types, especially their subtypes, is far from satisfactory. We expect that findings in this paper would help to provide insights into the RIII of human intestines. These new results and discussion have been added to the **Discussion** (page 21, lines 448–461) section in the revised manuscript.

Figure R3. (A) UMAP projection of 2,045 stem cells from Fawcner-Corbett, D. et al., colored by Seurat cluster identities. (B) Heatmap of the average expression of the selected ISC function-related marker genes in the five stem cell clusters. (C) UMAP projection of 811 monocytes and macrophages from Fawcner-Corbett, D. et al., colored by Seurat cluster identities. (D) Heatmap of the average expression of the selected macrophage function-related marker genes in the four macrophage clusters.

Finally, thank you again for your time and effort, and for helping to improve the manuscript. We have highlighted the changes in the revised manuscript. We hope that these changes have made it more appropriate for publication, and we look forward to your response.

REVIEWERS' COMMENTS:

Reviewer #1 (Remarks to the Author):

The revised manuscript addressed majority of the comments made in previous report.

Minor comment:

Previous report suggested that all mucosal resident macrophages are a continuum of newly recruited bone marrow derived monocytes (PMID: 22025201). Injury and/or inflammation modulates monocyte/macrophage homeostasis with the increase in inflammatory cell population and inhibition of differentiation to tissue resident phenotype. Observations author made may suggest this phenomenon as radiation induces local inflammation along with injury. Author needs to discuss that in the manuscript.

Reviewer #2 (Remarks to the Author):

In this revised manuscript, the authors have carefully revised the manuscript point by point with my comments. Taking into account the responses to all the other reviewers' comments, I think this version left nothing to be desired and will be acceptable to the journal.

Reviewer #3 (Remarks to the Author):

The authors have very well responded to the comments regarding the applications translating to humans and the technical details. Performing same analyses on human with such a large dataset is difficult. The regulators and their functions the authors mentioned in the discussion show the potential of the application of this study to be applied on humans. The authors have also upgraded there analysis tools to the latest version, which makes their analysis more reliable. The scRNA-seq QC steps are much more clear now, letting the work possible to be reproduced.

Revision of COMMSBIO-23-1149A

Title: Single-Cell Map of Dynamic Cellular Microenvironment of Radiation-Induced Intestinal Injury

Point by point reply to the reviewers.

We express our gratitude to the reviewers for your insightful comments. We have enclosed a revised version that will hopefully address the concerns raised by the reviewers. Specifically, we have incorporated a more thorough discussion regarding the balance between pro-inflammatory and resident macrophages in the context of inflammation induced by radiation. We detail our responses to the reviewers below.

In response to Reviewer #1, we would like to thank the reviewer for taking your time to evaluate our work. We are grateful for the positive feedback and addressed the issues as follows:

“The revised manuscript addressed majority of the comments made in previous report.”

Minor comment:

“Previous report suggested that all mucosal resident macrophages are a continuum of newly recruited bone marrow derived monocytes (PMID: 22025201). Injury and/or inflammation modulates monocyte/macrophage homeostasis with the increase in inflammatory cell population and inhibition of differentiation to tissue resident phenotype. Observations author made may suggest this phenomenon as radiation induces local inflammation along with injury. Author needs to discuss that in the manuscript.”

Response: We thank the reviewer for your valuable suggestion. As reported in the reference the reviewer mentioned, the macrophage pool in the intestine of adult mice requires constant replenishment from circulating monocytes and resident and inflammatory macrophages would represent alternative differentiation outcomes of the same precursor. In this study our data revealed the decrease of resident population and increase of inflammatory population in the radiation induced intestine injury and these results suggested that the bidirectional differentiation of peripheral *Ly6c*⁺ monocytes to pro-inflammatory macrophages or *Cx3cr1*⁺ resident macrophages may be important for balancing the radiation-induced inflammation and resident macrophages recovery. We have incorporated these discussion in the **Discussion** section (page 20, lines 419–424) in the revised manuscript accordingly.

Finally, thank you again for your time and effort, and for helping to improve the manuscript. We have highlighted the changes in the revised manuscript. We hope that these changes have made it more appropriate for publication, and we look forward to your response.